# Synthesis of Poly-γ-Glutamic Acid and Its Application in Biomedical Materials

**DOI:** 10.3390/ma17010015

**Published:** 2023-12-19

**Authors:** Minjian Cai, Yumin Han, Xianhong Zheng, Baigong Xue, Xinyao Zhang, Zulpya Mahmut, Yuda Wang, Biao Dong, Chunmei Zhang, Donghui Gao, Jiao Sun

**Affiliations:** 1Department of Cell Biology and Medical Genetics, College of Basic Medical Science, Jilin University, Changchun 130021, China; 2State Key Laboratory on Integrated Optoelectronics, College of Electronic Science and Engineering, Jilin University, Changchun 130012, China; dongb@jlu.edu.cn; 3Department of Anesthesiology and Operating Room, School and Hospital of Stomatology, Jilin University, Changchun 130012, China

**Keywords:** poly-γ-glutamic acid, preparation methods of γ-PGA, drug delivery, wound healing, tissue engineering

## Abstract

Poly-γ-glutamic acid (γ-PGA) is a natural polymer composed of glutamic acid monomer and it has garnered substantial attention in both the fields of material science and biomedicine. Its remarkable cell compatibility, degradability, and other advantageous characteristics have made it a vital component in the medical field. In this comprehensive review, we delve into the production methods, primary application forms, and medical applications of γ-PGA, drawing from numerous prior studies. Among the four production methods for PGA, microbial fermentation currently stands as the most widely employed. This method has seen various optimization strategies, which we summarize here. From drug delivery systems to tissue engineering and wound healing, γ-PGA’s versatility and unique properties have facilitated its successful integration into diverse medical applications, underlining its potential to enhance healthcare outcomes. The objective of this review is to establish a foundational knowledge base for further research in this field.

## 1. Introduction

At present, polymer materials find extensive applications in various industries such as medicine, agriculture, cosmetics, and more. Polyglutamic acid (PGA), a naturally occurring polymer comprising glutamic acid units, has gained significant recognition due to its inherent biocompatibility and degradability. PGA is synthesized by polymerizing D- and L-glutamic acid through amide bonds and can be classified into α-PGA and γ-PGA based on the type of amide bonds (Figure 1) [1]. The α-PGA is primarily manufactured through complex chemical processes, while γ-PGA can be easily produced on a large scale through biosynthetic methods. The initial discovery of γ-PGA was in the capsule of Bacillus anthracis, where it served to shield bacteria from the immune system [2]. Subsequently, researchers identified γ-PGA in Bacillus subtilis, Bacillus natto, and even some archaea bacteria. It exhibits three chemical stereostructures: γ-D-PGA, γ-L-PGA, and γ-D/L-PGA [3]. The molecular weight of γ-PGA varies depending on the ratio of D- and L-glutamic acid, ranging from 10 to 10,000 kDa. The applications of γ-PGA with different molecular weights also differ. For instance, low-molecular-mass γ-PGA is widely employed in pharmaceuticals and agriculture [4], while high-molecular-mass γ-PGA can serve as an additive to enhance calcium absorption [5].

The γ-PGA possesses a range of distinctive characteristics, including non-cytotoxicity, adsorption, water retention, and most notably, its biodegradability, which offers significant advantages for polymer material applications [6]. The physicochemical properties and applications of γ-PGA are intricately linked to its molecular structure. The presence of hydrogen bonds in γ-PGA imparts excellent water absorption properties, making it invaluable in wound dressing research. Additionally, the free carboxyl groups within γ-PGA can effectively bind metal ions, rendering it a promising carrier for addressing conditions related to copper toxicity and ferroptosis within the body. In 2001, Subramanian and colleagues made a seminal discovery by demonstrating that γ-PGA can establish a water environment through hydration repulsion with cationic lipid membranes, marking a pivotal advancement in drug delivery [7]. Xu and his team found that γ-PGA, with its exceptional water absorption capacity, exhibits anticoagulant properties, holding significant promise for future applications in conditions such as thrombosis [8]. As a result of its eco-friendly nature and outstanding properties, γ-PGA has emerged as a focal point of research, with its application domains continuously expanding. In the convergence of various fields, including materials science and biomedicine, polymer materials have proven to be powerful tools in addressing a multitude of medical challenges, offering potential solutions for numerous future healthcare issues.

While previous reviews on γ-PGA have primarily delved into its biosynthesis in microorganisms and its extensive applications in agriculture, this review takes a distinctive approach by concentrating on the synthesis methods and medical applications of γ-PGA. It offers an in-depth and comprehensive exploration rooted in recent research findings. By doing so, this review aims to serve as a valuable resource, providing substantial insights and practical guidance for scholars in the fields of materials science and medicine who are engaged in the quest for novel drugs or effective drug carriers for disease treatment.

## 2. Preparation Methods of γ-PGA

Various methods are employed to prepare γ-PGA, including microbial fermentation, chemical synthesis, and enzymatic polymerization. Microbial fermentation, utilizing bacteria such as Bacillus subtilis, offers an environmentally friendly and cost-effective approach to produce γ-PGA. Chemical synthesis involves the polymerization of glutamic acid monomers through amidation or esterification reactions. Enzymatic polymerization, catalyzed by enzymes like γ-glutamyl transpeptidase, presents a green and controlled method for PGA synthesis (Table 1).

### 2.1. Chemical Synthesis

Chemical synthesis methods include traditional peptide synthesis and dimer condensation polymerization. In traditional synthesis methods, the amino and carboxyl groups of glutamic acid are prone to form carboxylic anhydrides, which participate in the cyclization reaction. Therefore, the entire process needs to involve steps such as group protection and deprotection (Figure 2A) [9]. The synthesis process is relatively complex, and there are many by-products, making the purification process of the products more difficult. These things considered, more α-PGA is obtained through this method, and α-PGA degradation is more difficult, which can easily cause environmental pollution. In 2001, Sanda et al. optimized the chemical synthesis method and synthesized γ-PGA in the form of glutamic acid dimer, simplifying the synthesis process [10]. However, the production process can have a certain impact on the environment, this method has high cost and severe environmental pollution, making it rarely used for industrial production. Therefore, chemical synthesis methods do not have their own advantages in terms of synthesis efficiency, economic value, and environmental protection, but it is helpful for exploring its synthesis-related mechanisms and other issues.

### 2.2. Enzyme Conversion

Enzyme conversion refers to utilizing the glutamic acid monomers to carry out enzymatic reactions under the action of glutamine transpeptidase (GTP) (Figure 2B) [11]. The enzyme conversion method can avoid the negative feedback effect of product accumulation on production, and the obtained product is single, which can accumulate to obtain high-purity products. The key enzyme is mainly GTP, which can catalyze the transfer of glutamyl groups to receptors. When both the donor and receptor are glutamic acid, polyglutamic acid can be synthesized through automatic peptide conversion. Although this method is simple, the content of this enzyme is relatively low, making it very difficult to isolate and purify this enzyme, and this enzyme can only exhibit γ-PGA synthesis activity on the cell membrane. Hence, this greatly limits the application of enzyme conversion methods, and finding easily available enzymes that can replace GTP may eliminate this limitation. In 2011, Kino et al. discovered a ribosomal protein S6 modifying enzyme called RimK in Escherichia coli that can promote the synthesis of γ-PGA from L-glutamic acid [12]. It has strict substrate specificity and good heat resistance, so it can be used in industrial production. However, due to the quality of γ-PGA, Glu, and other mixtures obtained by the precipitation method, the specific yield cannot be measured. If a specific method of measurement can be found, RimK may be able to replace GTP.

### 2.3. Extraction

Natto has stickiness, and its source of stickiness is γ-PGA. The extraction method generally refers to the method of extracting γ-PGA from the natto using organic solvents. (Figure 2C) The process mainly involves extracting γ-PGA in water by streaming and soaking natto and further extracting it through organic solvents. This method is simple and easy to operate, so extraction is the main method to obtain γ-PGA in the early stage. However, natto is mainly a mixture of fructose and γ-PGA, and the content of γ-PGA is relatively low. Simply using organic solvents often results in a crude product of γ-PGA with low yield. Accordingly, the extraction method cannot be applied to large-scale production of γ-PGA.

### 2.4. Microbial Fermentation

Microbial production of γ-PGA stands as the predominant method in industrial production, characterized by its environmental friendliness and scalability. Commonly employed strains include B. subtilis and B. licheniformis, among others. Presently, two main fermentation methods are utilized: solid fermentation [13] and liquid fermentation [14]. Liquid fermentation has gained prominence in industrial production due to its advantages over solid-state fermentation. Solid fermentation often suffers from uneven nutrient distribution, limiting the application of biosensors and fine-tuning of the fermentation process. On the contrary, liquid fermentation offers effective control of key fermentation parameters such as temperature, pH, and dissolved oxygen, while enabling real-time monitoring. This makes it an economical and practical choice for microbial fermentation. However, as γ-PGA accumulates in the liquid fermentation process, challenges emerge in the form of increased broth viscosity and bubble formation. Real-time monitoring of broth concentration and aseptic operations become crucial. Moreover, optimizing the fermentation process is essential to address the issue of low yield.

Strains responsible for γ-PGA production can be categorized into two types: glutamate-dependent and glutamate-independent strains, each following a different synthesis process. Glutamate-independent strains generate the required glutamic acid internally during synthesis, starting with the strain converting the carbon source via the glycolysis pathway and TCA cycle into α-ketoglutarate. The α-ketoglutarate is then transformed into L-glutamate via glutamate dehydrogenase, followed by racemization into D-glutamate. Ultimately, the strain employs PGA synthase to convert L-glutamate and D-glutamate into γ-PGA (Figure 2D). In contrast, glutamate-dependent strains necessitate the addition of exogenous glutamate and begin their synthesis process directly from the racemization step.

Currently, there is a wealth of research on glutamate-dependent strains, including notable examples such as B. subtilis D7 [15], B. subtilis MJ80 [16], ZJU-7 [17], and Bacillus tequilensis BL01 [18]. However, studies on glutamate-independent strains are relatively limited. In 2017, Zeng and colleagues isolated a heat-resistant glutamate-independent strain, B. subtilis GXG-5 [19], which exhibited optimal production performance at 50 ℃. Under optimal fermentation conditions, B. subtilis GXG-5 outperformed B. subtilis GXA-28, a glutamate-dependent strain, in terms of yield and substrate utilization. Generally, the yield of glutamate-independent strains tends to be lower than that of glutamate-dependent strains. However, their production costs are significantly reduced as they do not require the addition of exogenous glutamate.

## 3. Optimize the Production of γ-PGA

Optimizing γ-PGA production can be broadly categorized into two main approaches. The first method involves fine-tuning the fermentation process to increase the synthesis of critical components, leading to enhanced production. The second approach focuses on the fundamental transformation of microorganism metabolism by modifying their genetic characteristics through genetic and metabolic engineering. The former method can rapidly boost yield by improving fermentation conditions, while the latter necessitates the construction of plasmids, breeding, and other processes, requiring more time but yielding new strains that can be stably inherited.

### 3.1. Metabolic Process Regulation

Following the identification of high-yield strains, optimization of fermentation conditions (e.g., pH, temperature), culture media (carbon and nitrogen sources), and the selection of appropriate fermentation methods (e.g., feeding strategies) are typically employed to boost production. Through these avenues, production rates can be rapidly enhanced. Recent studies have indicated that the addition of specific ions during the production process can significantly stimulate γ-PGA synthesis. For instance, in 2023, Guo et al. discovered that the introduction of exogenous iron ions can elevate γ-PGA production levels in Bacillus licheniformis CGMCC (Figure 3) [20]. The experiment first determined the optimal concentrations of key fermentation components in a shaker, which included untreated sugarcane molasses (9% soluble solids) without nitrogen supplementation (4 g/L yeast extract), 0.7 g/L FeSO_4_·7H_2_O, and 80 g/L monosodium glutamate. Subsequently, production was conducted in a 5 L bioreactor, and the γ-PGA yield in the group with exogenous ions reached 70.436 g/L, while the blank control group only achieved 40.668 g/L. After further optimization of agitation and aeration, the final yield reached 76.848 g/L, presenting an economical route for γ-PGA synthesis. Researchers have also explored the underlying mechanisms. However, it is worth noting that the divalent iron ions used in the experiment are susceptible to oxidation into trivalent iron, which raises the challenge of maintaining divalent iron in its reduced state during production. Additionally, the addition of Na^+^, Ca^2+^, and Mn^2+^ [21,22] has all been found to enhance γ-PGA production, though the specific mechanisms remain to be fully elucidated.

Moreover, utilizing industrial or agricultural waste as fermentation substrates not only promotes production to some extent but also aligns with environmentally friendly practices, making it a green and cost-effective optimization method. In 2011, Yong et al. [23] demonstrated that using cow manure as a substrate for γ-PGA production resulted in a yield of 0.0437 g/L of product per gram of substrate at 37 °C for 48 h. In 2015, Tang et al. utilized rice straw as a substrate, achieving increased yield while reducing carbon source consumption. In 2020, wastewater from yeast molasses fermentation was repurposed to produce fulvic acid (FA) for PGA production [24]. This approach not only replaces low-cost substrates with inexpensive FA but also minimizes environmental pollution associated with yeast molasses fermentation. Traditionally, sucrose was the primary substrate, but now non-food raw materials are being used, reducing costs and benefiting the environment. This dual impact makes it both an economically and environmentally friendly optimization method.

While these methods have indeed increased γ-PGA production to some extent, they may not fully address the limitations imposed by the strains themselves. To secure stable, high-yield strains, it is necessary to genetically modify the strains to engineer bacteria suitable for industrial production.

### 3.2. Transforming Bacterial Strains

The genes responsible for γ-PGA synthesis can be categorized based on the state of γ-PGA: capsule (cap) genes, if the synthesized γ-PGA binds to the cell wall, and pgs genes if it is released into the environment. In 2005, Candela et al. discovered that cap B, cap C, cap A, and cap E were essential for γ-PGA synthesis. In 1999, Ashiuchi et al. [25] identified that in Bacillus subtilis, three genes were involved in γ-PGA synthesis, namely pgs B, pgs C, and pgs A (also known as yws C, ywt A, and ywt B). In 2005, Candela et al. [26]. found that in addition to the three previously mentioned genes, pgs E (also known as ywt C), owing to its high homology with capE, is highly likely to be involved in γ-PGA synthesis. The pgd gene, situated downstream of the pgs operon, encodes the γ-PGA degradation enzyme. In 2011, Cao et al. successfully cloned the γ-PGA synthase gene pgs BCA from Bacillus amyloliquefaciens LL3 and expressed it in Escherichia coli JM109 using the pTrclpgs vector, resulting in increased γ-PGA production. Notably, pgs BCA originates from a glutamate-independent strain, making it advantageous for mechanistic and production applications [27]. This leads to the question of whether we can significantly enhance γ-PGA production by cloning the synthase genes from high-yield strains and introducing them into strains with higher glutamic acid levels.

Furthermore, the gene responsible for γ-PGA degradation also plays a pivotal role in the synthesis of γ-PGA. In 2013, Scoffone et al. deleted the γ-PGA degrading enzyme genes pgd S and ggt in B. subtilis 168, resulting in a substantial increase in γ-PGA production. Knocking out these two genes enabled the experimental group to achieve yields on par with the wild-type strains, facilitating research convenience [28]. In 2019, Yoshihiro et al. obtained Bacillus licheniformis RK14 from soil and induced mutations through ethyl methanesulfonate (EMS) treatment to obtain a high-yield strain, RK14-46 [29]. The introduction of plasmids was used to disrupt the pgd S and ggt genes. Experiments revealed that the deletion of the pgd S gene hindered γ-PGA synthesis; whereas, the removal of the ggt gene increased γ-PGA production, resulting in an elevated molecular weight of γ-PGA. The impact of pgd S varies among different strains, emphasizing the need to consider whether a gene has a positive or negative influence on synthesis during production optimization.

Recent studies have shed light on gene and enzyme regulation in metabolic processes. In 2021, Li et al. achieved a remarkable 62% increase in γ-PGA production, reaching 12.02 g/L using Bacillus licheniformis [30]. This involved the overexpression of pyruvate dehydrogenase (Pdh ABCD) and citrate synthase (Cit A), the inhibition of isocitrate lyase (ace A), and the deletion of the pyruvate formate-lyase gene pfLB. In 2022, Zhu et al. harnessed crude glycerol as a substrate and introduced robust promoters to boost the expression of synthesis-related genes while simultaneously eliminating pathways producing by-products [31] (Figure 3). As a result, γ-PGA production by Bacillus amyloliquefaciens soared to 26.4 g/L, a remarkable 3.72-fold increase compared to the original strains. These regulatory steps primarily centered on the tricarboxylic acid (TCA) cycle, underscoring the crucial role of TCA in the PGA synthesis process and highlighting the importance of oxygen supply during fermentation.

**Figure 3 materials-17-00015-f003:**
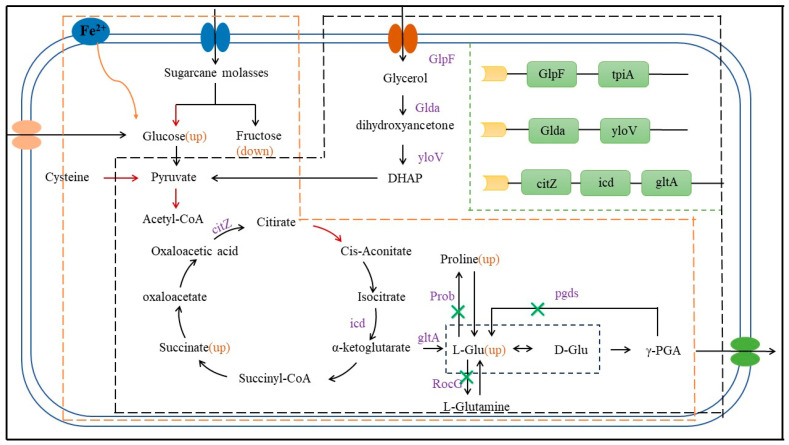
Regulation of γ-PGA synthesis via Fe^2+^/genetic engineering. The yellow box represents the regulation of Fe^2+^on synthesis, and the red arrows indicate a promoting effect on the processes [20]. The black box indicates regulation through genetic engineering, with the green box indicating the insertion position of the strong promoter and the green crosses indicating the deletion of this metabolic pathway by knocking out relevant genes [31].

Furthermore, optimizing bacteria to utilize cost-effective substrates through metabolic engineering is a key strategy. Sucrose, as an abundant and affordable substrate, can substantially reduce γ-PGA production costs [32]. In an experiment, the native sucrose utilization pathway of Bacillus amyloliquefaciens NK-1 was removed, and new utilization pathways were introduced. The outcome demonstrated a 49.4% increase in sucrose consumption and a 38.5% boost in γ-PGA production compared to the initial process. By implementing novel substrate utilization pathways through metabolic engineering, we can efficiently produce γ-PGA using more economical substrates, ultimately lowering production costs and facilitating the commercialization of γ-PGA.

It is important to note that while genetic and metabolic engineering can significantly enhance the yield of the original strain, achieving exceedingly high yields remains a challenge. For future production optimization, a combination of genetic and metabolic modifications may fully harness the strengths of both approaches, offering prospects for improved production efficiency. Continued exploration of the synthesis mechanisms within strains is equally crucial, as it can lead to the identification of new targets for enhancing yield. Concurrently, ongoing efforts to identify and obtain new high-yield strains will greatly contribute to future endeavors.

In comparison to the other three methods, microbial fermentation presents unparalleled advantages. Nevertheless, there are substantial challenges in its production process, such as strains activating degradation pathways during synthesis, hindering product accumulation. As products accumulate, the viscosity of the fermentation broth increases, making separation and purification challenging. Thus, the development of high-quality engineered strains through genetic and metabolic engineering will play a pivotal role in addressing these challenges and advancing microbial fermentation.

## 4. The γ-PGA Hydrogel

The γ-PGA often appears in the form of hydrogel in the application process [33]. Hydrogel is a kind of hydrophilic polymer material, which has the ability to absorb and retain water. The preparation of hydrogel is mainly divided into the physical crosslinking method and chemical crosslinking method. The physical crosslinking method is mainly to form hydrogel through the interaction between molecules. Due to the lack of crosslinking agent, it reduces the toxicity to cells and shows good biocompatibility. The hydrogel prepared using the chemical crosslinking method has good mechanical strength and longer degradation time. In addition, γ-PGA hydrogel belongs to natural polymer hydrogel, which has good biocompatibility and low cytotoxicity, and there are a lot of free carboxyl groups in γ-PGA, providing good water absorption for hydrogel. However, when the water content of γ-PGA is high, its mechanical capacity will decline, so different materials are often added to prepare composite hydrogels to improve its shortcomings.

### 4.1. Crosslinking with Chitosan

Chitosan (CS) is also a naturally occurring polymer with water retention, diffusivity, and antibacterial properties. However, its mechanical strength is poor and its degradation time in vivo is difficult to control, as well as its poor hydrophilicity and cell compatibility, which have caused difficulties in its application. In 2006, Kang et al. prepared polyelectrolyte complex (PEC) hydrogels using γ-PGA as a polyanion electrolyte and CS as a polycationic electrolyte using freeze-drying technology [34]. Through comprehensive characterization, it has been established that PEC hydrogel exhibits notable attributes such as heightened tensile strength and a more compact porous structure, which, in turn, fosters the proliferation of normal human dermal fibroblast (NHDF) cells. Moreover, it has been confirmed that the hydrogel’s swelling ratio can be altered by adjusting pH levels. The development of pH-sensitive hydrogels holds substantial promise in the medical field. For instance, it enables the initiation of drug release based on self-swelling triggered by variations in pH within different organs. When applied to wound dressings, it can promote wound healing by responding to the wound’s pH conditions. Nie et al. prepared CS/γ-PGA hydrogel using the electrostatic contact method in 2020 (Figure 4A) [35]. The process includes adding CS and γ-PGA into NaOH to dissolve, adding pH regulator to adjust pH, and finally preparing hydrogel using the freeze-drying method. During the experiment, it was found that different pH regulators, γ-PGA contents, and drying methods had an impact on the porosity of the hydrogel. The hydrogel prepared in this way had a high swelling rate and high pH sensitivity, which can be used as potential materials in future drug delivery and other applications.

The PEC hydrogel initially suffers from an uneven gel network and limited toughness. However, these limitations can be effectively addressed by modifying the physical and chemical properties and functions of CS/γ-PGA hydrogel. In 2022, Wang et al. explored a multi-step non-covalent crosslinking strategy for CS/γ-PGA hydrogel by incorporating Dendrobium candidum enzyme (DOE) (Figure 4B) [36]. They discovered that an increase in γ-PGA content and the addition of DOE resulted in a hydrogel with higher porosity, significantly enhancing its hydrophilicity and water retention. Interestingly, the experiment revealed that an excessive or insufficient amount of DOEs did not improve the hydrogel’s tensile properties. Instead, the key to preparing CS/γ-PGA/DOE hydrogels lay in maintaining a concentration of DOEs between 4 and 6%. Notably, these hydrogels displayed the ability to inhibit the growth of Escherichia coli and Staphylococcus aureus, while low concentrations of DOEs were found to promote cell proliferation. With DOE’s free radical scavenging properties, this hydrogel holds great potential for applications in skincare, medicine, aesthetics, and addressing oxidative stress-related diseases in the future. The charge density of chitosan is known to be pH-dependent, suggesting the possibility of creating pH-sensitive hydrogels with enhanced performance through crosslinking with γ-PGA [37,38]. Currently, CS/γ-PGA hydrogels have gained significant attention in the field of drug delivery, wound dressing, and other applications. They provide a promising alternative hydrogel material for advancements in the medical field.

### 4.2. Crosslinking with Hyaluronic Acid

Hyaluronic acid (HA) is a natural polysaccharide and one of the components of ECM [39]. It has been widely used in various ophthalmic surgeries, can accelerate wound healing, and has broad prospects in the cosmetics industry. However, pure HA hydrogel is hard and degrades quickly. Research has found that γ-PGA can improve the shear resistance of HA. HA/γ-PGA hydrogel has a rich porous structure [40]. Therefore, this hydrogel is a good material to provide a potential therapeutic method for future medical applications.

Presently, researchers have made a series of modifications to hydrogels to enhance their physical and chemical properties. In 2018, Ma et al. developed an injectable hydrogel through the photopolymerization of methacrylate-functionalized HA and γ-PGA (Figure 4C) [41]. This hydrogel exhibited superior load-bearing capacity and toughness compared to conventional hydrogels. The preparation process involved dissolving HA in deionized water for 12 h, adding glycidyl methacrylate (GMA) with a molar ratio of GMA: -COOH at 3:1, adjusting the pH to 4.5~5.0, and heating it at 60 °C for 6 h. After 72 h of dialysis, HA-GMA was prepared using a freeze-drying method. The preparation process of γ-PGA-GMA was similar, with the only difference being the molar ratio of GMA: -COOH at 0.5:1. The two components were dissolved in PBS and crosslinked under ultraviolet (UV) light irradiation to obtain the hydrogel. With an increase in γ-PGA content, the hydrogel displayed increased swelling ratio and decreased compression modulus, indicating improved water absorption and reduced rigidity of the hydrogel. In vitro drug release experiments with BSA-loaded hydrogel showed sustained drug release over 5 days, with a cell survival rate of over 90% for NIH3T3 mouse fibroblasts. The hydrogel exhibited excellent cell compatibility and shape recovery ability and holds great promise as an injectable drug carrier.

Adaptive hydrogels often suffer from weak mechanical strength and rapid degradation [42]. In 2020, Ma et al. developed an HA/γ-PGA-adaptive hydrogel using a combination of dynamic covalent chemistry, stable covalent chemistry, and interpenetrating polymer network (IPN) strategy (Figure 4D) [43]. Initially, aldehyde HA (HA-CHO) and hydrazide-functionalized γ-PGA (γ-PGA-ADH) were crosslinked via Schiff base reactions to form the first layer of the network structure. The γ-PGA-GMA was photopolymerized to create the second layer of the network structure, and finally, the hydrogel was formed through UV irradiation. This method produced a hydrogel with a stable structure, good mechanical properties, enzyme degradability, and injectability. The hydrogel had the ability to rapidly gel and ensured it would not spread beyond the injection site. However, the swelling behavior of the hydrogel could pose challenges for drug diffusion. Drug release experiments were conducted with BSA loading, revealing that the hydrogel could release 10% of the drug after 60 h and continue to release for 72 h. This hydrogel holds significant potential for medical applications in weight-bearing tissue engineering.

Currently, HA/γ-PGA hydrogel is primarily applied in drug delivery, but its potential extends to tissue engineering and various other industries for the future.

**Figure 4 materials-17-00015-f004:**
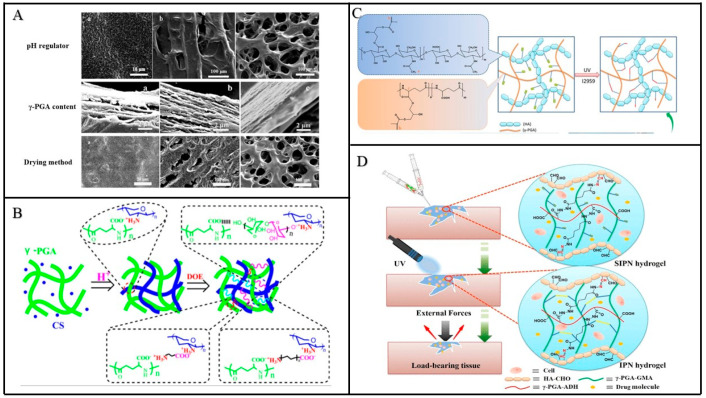
(**A**) The porous structure of hydrogels formed by different PH regulators, different γ-PGA contents, and different drying methods [35]. (**B**) Effect of DOEs on CS/γ-PGA hydrogel [36]. (**C**) Preparation principle of HA/γ-PGA [41]. (**D**) HA/γ-PGA hydrogel based on dynamic covalent chemistry, stable covalent chemistry, and IPN strategy [43].

### 4.3. Crosslinking with Gelatin

Gelatin (GEL) is a natural polymer material with good film-forming and degradability and is widely used to study the regeneration of tubular viscera [44,45]. However, the mechanical properties and thermal stability of GEL are poor, and it is usually necessary to crosslink with other molecules when preparing hydrogel. GEL/γ-PGA hydrogel has demonstrated its effectiveness in repairing intervertebral disc injuries [46,47]. The preparation process involves crosslinking GEL and γ-PGA using 1-(3-dimethylaminopropyl)-3-ethyl-carbodiimide (EDC) in specific proportions to form a hydrogel. The incorporation of γ-PGA enhances the mechanical strength of the hydrogel, effectively addressing GEL’s inherent limitations. Recently, Dou et al. introduced physically crosslinked GEL hydrogel into covalently crosslinked γ-PGA hydrogel, creating a double-network hydrogel (Figure 5A) [48]. The porous structure of this hydrogel becomes more uniform with increasing GEL content. This might be due to the low concentration of γ-PGA reducing the covalent crosslinking density, slightly increasing the pore size of the hydrogel. However, excessively high GEL content can disrupt the gel structure. Moreover, hydrogels prepared through this method exhibit remarkable thermal stability and excellent mechanical properties. It is noteworthy that the hydrogel maintains a light transmittance of over 70%, which is crucial for wound observation when used as a wound dressing [49,50]. As a result, GEL/γ-PGA stands as a promising material for wound dressings and provides a biocompatible resource for medicinal development.

### 4.4. Crosslinking with Other Materials

Collagen has been widely used in the beauty industry at present, while collagen/γ-PGA hydrogel is rarely studied. Concentrated collagen has the problem of high viscosity, while low-concentration collagen has poor mechanical strength. In 2017, Cho et al. found that the introduction pf γ-PGA into concentrated collagen can reduce the viscosity, which can be used to prepare injectable hydrogel (Figure 5B) [51]. Sodium alginate (SA) has the advantages of good gel formation, low toxicity, and being able to load cells [52,53], but it is easy to degrade under conditions of high temperature, UV, and oxidant, so a mild method should be used to prepare hydrogel. Wang et al. prepared SA/γ-PGA hydrogel through introducing microcrystalline cellulose (MCC), which contains a multi-hydrogen bond structure (Figure 5C) [54], improving the mechanical strength of the hydrogel. MCC is introduced into SA-CHO and then dissolved with γ-PGA-ADH in PBS to prepare hydrogel, prolonging its degradation time, thus avoiding the problem of fast degradation of γ-PGA and SA. At present, many researchers are still searching for other suitable materials for crosslinking with γ-PGA and have achieved certain achievements.

**Figure 5 materials-17-00015-f005:**
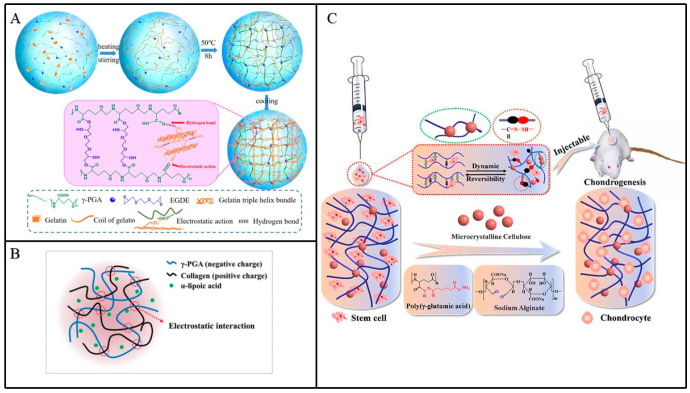
(**A**) Schematic diagram of GEL/PGA hydrogel preparation [48]. (**B**) Schematic diagram of collagen/PGA hydrogel crosslinking [51]. (**C**) A method to stabilize the SA/γ-PGA hydrogel network by adding MCC, and research has found that it can promote the development of stem cells into chondrocytes [54].

## 5. PGA-Based Drug Delivery Systems

PGA possesses a high loading capacity for both hydrophobic and hydrophilic drugs, making it an ideal candidate for drug delivery systems. PGA-based nanoparticles, microparticles, and hydrogels offer controlled and sustained drug release profiles.

### 5.1. PGA-Based Anti-Tumor Drug Delivery

PGA-based nano-carrier can specifically adhere to β-glutamyl transferase on tumor cell membranes, so they can be used to deliver chemotherapy drugs such as paclitaxel (PTX) Adriamycin (DOX) to enhance their targeting [55]. As early as 1998, Li et al. formed a PTX–PGA conjugate using PTX covalent bond with PGA. This PTX–PGA conjugate was found to have better anti-tumor activity than ordinary PTX in mouse tumor models [56]. In 2005, Singer synthesized a paclitaxel polylumex from PTX and PGA and found that this conjugate has a slowing effect on PTX [57]. Arroyo-Crespo et al. used PGA to connect DOX and endocrine drugs (aromatase inhibitor aminoglutethimide) to neutralize excess reactive oxygen species (ROS) produced by DOX during treatment and achieve synergistic therapeutic effect (Figure 6A–C) [58]. In connection with previous studies, for prostate cancer cells treated with DOX in combination with sildenafil, sildenafil can produce nitric oxide to neutralize ROS produced by DOX [59]. Therefore, the synergistic administration of these two drugs, by linking them with PGA, may be a new direction for the treatment of prostate cancer. Nisar et al. prepared glutamate-grafted chitosan (CH-g-GA) hydrogel microbeads using gamma-irradiated chitosan and L-glutamate to control the release of DOX. It was observed that the pH triggering the release of DOX was 5.8, closely mirroring the pH of the tumor microenvironment. Consequently, the hydrogel demonstrates the capability to precisely deliver the drug to the specific location of the cancer [60]. Guo et al. used α-PGA/DOX nanoparticles to detect the anti-tumor activity of 4T1 cells and found that α-PGA-DOX NPs enhanced the anti-tumor efficacy in vivo, and the tumor inhibition rate was 67.4%, which was 1.5 times higher than that of DOX injection [61].

PGA demonstrates versatile capabilities in augmenting the effectiveness of various chemotherapeutic drugs. It can establish covalent bonds with small-molecule chemotherapeutic drugs and, in the case of cisplatin (CDDP), engage in chelation, consequently enhancing the anti-tumor efficacy of CDDP [62]. Li et al. developed a camptothecin (CPTP)-coupling prodrug (CPT) micelle using block copolymer mPEG-PGAp with varying glutamate (GA) repeat units (20, 40, and 60). These were synthesized via BLU-NCA ring-opening polymerization, with active methoxy polyglycol amine (mPEG-NH_2_) as the initiator. The CPTP/CDDP precursor micelle was established through the mutual coordination of CDDP (Pt)-COOH chelates. Notably, CDDP enhanced the stability of CPTP (Figure 7B) [63]. Zhang et al. created a PGA-ASP maleimide–cisplatin peptide complex (PAMCP) loaded with CDDP, which was conjugated to a transferrin receptor (TFR) targeting peptide through maleimide functional splicing. MTT assays demonstrated that PAMCP exhibited targeted toxicity. In in vivo toxicity studies, PAMCP reduced the toxicity of CDDP while inhibiting tumor cell growth (Figure 7D) [64]. Wang et al. developed a PGA-platinum (IV) predrug nanoconjugated compound (γ-PGA-CA-Pt(IV)). This compound could rapidly convert into active platinum in response to the tumor microenvironment’s pH, exerting an anti-tumor effect while reducing the potential for off-target drug release and minimizing side effects (Figure 7A) [65]. Jiang et al. designed two PEG-separable L-PGA-cisplatin (PLG-Pt) nanocomplexes that responded to the acidic tumor microenvironment and the overexpressed matrix metalloproteinase-2/9 (MMP-2/9) within it. In the tumor microenvironment, the overexpression of MMP severed the bridging chemical bond between PEG and PLG. The released PLG-Pt exhibited enhanced drug delivery capabilities and increased anti-tumor efficacy. In comparison to non-separable PEG-PLG-Pt, the separable PEG-enabled nano-agents with tumor microenvironment responsiveness showed improved anti-tumor efficacy in HGSOC tumor models (Figure 7C) [66].

**Figure 7 materials-17-00015-f007:**
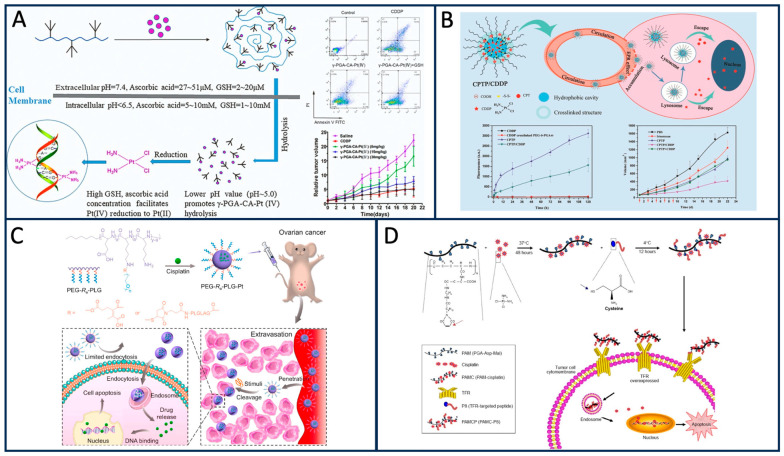
(**A**) The γ-PGA-CA-Pt(IV) was converted to Pt(II) in acidic environment, and the killing rate of Pt(IV) was enhanced after pre-incubation with GSH [65]. (**B**) CPTP/CDDP release CPT diagram [63]. (**C**) Schematic diagram of enhancing anti-tumor effect of PLG-CDDP nano-agent [66]. (**D**) Schematic illustration of cisplatin (CDDP)-loaded and TFR-targeted drug delivery systems constructed via the self-assembly method [64].

PGA demonstrates remarkable potential not only in delivering small molecular chemotherapy drugs but also in transporting large molecular anti-tumor substances. In 2023, Liu et al. harnessed the power of “click chemistry” involving azides and DBCO, as well as the high affinity of the FC-II-4C peptide for monoclonal antibody crystalline fragments (Fc). They used PGA to expand the number of drug-binding sites, resulting in the creation of eight antibody–polymer–drug couplers (APDCs). The impact of DAR on APDCs was subsequently assessed in SKOV-3 and MC38 tumor models, revealing that an increase in DAR enhanced the tumor-suppressive effect of APDCs. This experiment demonstrated that PGA, serving as an intermediary for coupling antibodies and drugs in APDC, had a positive effect. Notably, the DAR of MMAE reached 41.6, effectively overcoming the challenge of the low tolerance dose of MMAE in the human body. Nevertheless, the biochemical effects induced by PGA in this process will need further confirmation through subsequent experiments (Figure 8B) [67].

The method of chemical coupling of lysine with aspartame (Asp) to overcome the high immunogenicity of Asp has been applied in clinic. However, this method is limited by the number of lysine layers, so the shielding effect is poor. Yuan et al. synthesized a zymphoionic cloak with Asp coupling using multiple layers of short alternating glutamic acid and lysine (EK). This method can overcome the defect of lysine chemical coupling, and the multi-layer ion cloak provides better shielding effect. However, the disadvantages of this method are that the synthesis process is complex, it is high cost, and lack of reproducibility, which leads to clinical application difficulties, so it can be explored in this direction in the future (Figure 8C) [68].

At the cellular level, PGA has demonstrated its significance in the realm of cellular immunotherapy. Possessing excellent biocompatibility and biodegradability, PGA’s negative energy on the surface reduces the surface charge of nanoparticles, providing an effective shield for nanoparticle applications. This quality makes it highly suitable for the fabrication of nanomaterials utilized in chimeric antigen receptor T cell (CAR-T) preparations. Smith et al. employed PGA for the synthesis of a nanomaterial capable of in vivo T cell binding. This material facilitates the formation of CAR-T cells endowed with tumor-killing capabilities. The in situ CAR-T approach addresses cost issues associated with the traditional method of editing T cells in vitro and subsequently transfusing them back into the body. However, certain drawbacks persist, primarily concerning the specificity and toxicity of nanoparticles. Additionally, the co-expression of transposase genes is required for the effective encapsulation of CAR-expressing DNA fragments within nanomaterials (Figure 8A) [69].

**Figure 8 materials-17-00015-f008:**
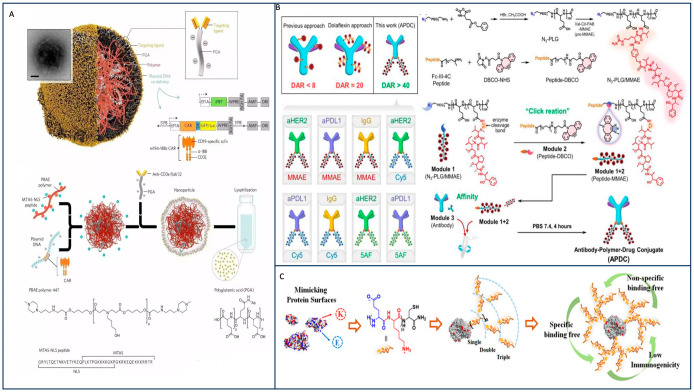
(**A**) Schematic diagram of assembling nanoparticles capable of producing CAR in situ [69]. (**B**) Various antibody–polymer–drug conjugates (APDCs) were synthesized through the use of polymer linkers, resulting in a “Lego-like” assembly approach [67]. (**C**) Schematic diagram of a zwitterionic cloak of multiple layers of short alternating glutamic and lysine (EK) peptides [68].

### 5.2. Delivery of Other Classes of Drugs Based on PGA

PGA is garnering increasing attention for its potential in anti-infection and anti-inflammatory applications. Nieto-Orellana et al. employed vitamin B12 ligands to bind to PGA copolymers, specifically targeting PEG Mosaic and therapeutic proteins. Initially, they demonstrated that Calu-3 cells in their epithelial model expressed the vitamin B12 internalizing receptor (CD 320). Subsequently, when lysozyme protein drug was introduced into the targeted PEG patch PGA copolymer as a dry powder, the cell internalization was found to be 2–3 times higher than that of the non-targeted copolymer. Importantly, this process did not induce a complement-activated immune response, possibly attributed to the negatively charged shielding effect of PGA (Figure 9A) [70]. Enshaei et al. synthesized a hydrogel/nanoparticle-loaded system using γ-PGA and poly(3, 4-ethylenedioxythiophene) (PEDOT) to control curcumin (CUR) release and compared the release curves of PEDOT/CUR and γ-PGA/PEDOT/CUR systems with and without electrical stimulation. It was found that γ-PGA could regulate the release of CUR via electrical stimulation [71].

PGA possesses pH-responsive properties that have been leveraged in various drug delivery systems: Das et al. employed 3-amino-propyl triethoxysilane (APTES) and a polyglutamine-based nanomaterial, specifically third-generation silicon network mesoporous bioactive glass (MBAG), to construct a system for the controlled release of erythromycin. This system was encapsulated in core-shell pH-responsive poly-L-amino acid-embedded microspheres (MBAG/PLGA) [72]. Chen et al. achieved pH-responsive drug delivery by dispersing chitosan (CS) powder uniformly in a solution of PGA and alginate (SA). Exposure to a gaseous acidic atmosphere allowed CS to gradually dissolve and interact with PGA and SA, forming a CS/PGA/SA–PEC complex hydrogel to control the release of Piroxicam (PXC) (Figure 9C) [73]. Wang et al. embedded probiotics into γ-PGA microgels to create nitric oxide (NO)-responsive delivery microcarriers. They constructed an NO-responsive γ-PGA hydrogel microcapsule (NRPM) using a microfluidic technology platform, which enabled precise production of uniform microspheres with a narrow size distribution (100–600 µm). The microgel system, based on double-bond functionalized γ-PGA, was developed through NO-reactive crosslinking agents and visible-induced cytocompatible crosslinking reactions. It achieved high-density embedding of lactic acid bacteria (LAB) and remained stable in a gastric acid environment. The system was designed to rapidly release LAB in response to NO molecular stimulation to effectively slow down colitis treatment (Figure 9B) [74].

**Figure 9 materials-17-00015-f009:**
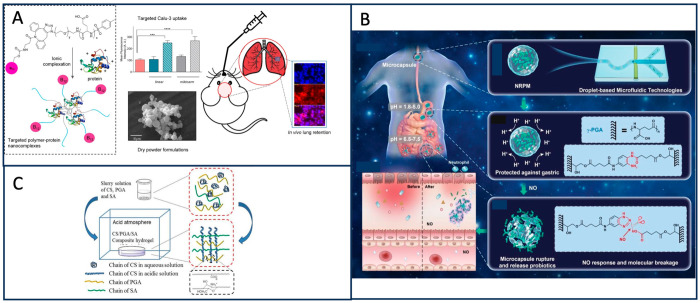
(**A**) The complexation reaction forms targeted polymer–protein particles, and mouse inhaled the drug (*** *p* < 0.001, **** *p* < 0.0001) [70]. (**B**) The production process and design schematic diagram of the new brushless torque method are presented [74]. (**C**) Shape comparison of the CS/PGA/SA hydrogel [73].

## 6. Tissue Engineering with PGA

The γ-PGA often acts as a biological scaffold in tissue engineering, which plays the role of extracellular matrix (ECM) for cell adhesion, proliferation, and migration. The γ-PGA scaffolds have shown promise in regenerating tissues such as bone, cartilage, and skin. Currently, most research is focused on cartilage tissue repair.

### 6.1. Cartilage Tissue Engineering

Due to the lack of blood vessels and lymphatic vessels in the cartilage, the migration ability of chondrocytes is weak, making it extremely difficult for cartilage damage to self-repair [75]. The γ-PGA hydrogel has become the focus of cartilage tissue engineering due to its high water content and porosity, characteristics similar to natural cartilage tissue, and ability to maintain the phenotype of chondrocytes. In 2020, Yang et al. (Figure 10A) [76] prepared methacrylate-γ-PGA (γ-PGA-GMA) and cysteamine-functionalized γ-PGA (γ-PGA-SH) hydrogels and implanted them into the ears of rabbits with cartilage injury, which can help cartilage reconstruction at the injured site. This type of gel boasts mild preparation conditions, commendable mechanical strength, and excellent degradation performance. However, in certain regions of the hydrogel, there exists a thick and irregular pore structure, and the porous architecture plays a crucial role in cell attachment and other functions [77]. Research indicates that addressing this defect can be achieved by controlling the content of γ-PGA, emphasizing the significance of exploring the specific γ-PGA content for optimal hydrogel performance. Additionally, hyaluronic acid (HA) and γ-PGA-crosslinked hydrogels have garnered significant attention. Studies confirm that HA-modified γ-PGA-GMA can enhance swelling behavior and mechanical properties while exhibiting superior cell compatibility [78]. This underscores the feasibility and promise of combining γ-PGA and HA for applications in tissue engineering. In 2021, Ma et al. (Figure 10B) [79] obtained dual crosslinked HA/γ-PGA hydrogels through Schiff base reaction and UV irradiation, and compared with single-crosslinked hydrogels, they have a longer degradation time. Because cartilage tissue repair time is longer, many HA/γ-PGA-crosslinked hydrogels have a fast degradation time, and often before repair, the hydrogel scaffold disappears, so it provides a new idea for solving this problem. From the vitro experiment, it was found that the hydrogel had good biocompatibility, but there was no specific in vivo experiment to verify it. These things considered, chitosan/γ-PGA hydrogel has also been widely used in regenerating cartilage [80]; on the basis of chitosan/γ-PGA hydrogel, the surface was modified by elastin, human serum protein (HSA), and poly-lysine, and the apoptosis was reduced after modification. However, no matter which kind of hydrogel is used, it should ensure that the hydrogel is non-toxic to cells and has good physical and chemical properties. It is also worth contemplating how to actualize the clinical application of γ-PGA hydrogel.

### 6.2. Neural Tissue Engineering

The γ-PGA has also been studied in neural tissue engineering [81]. Neurological disorders such as Alzheimer’s disease, Parkinson’s disease, and stroke present significant challenges to healthcare. Addressing these challenges involves the potential use of biological scaffolds for implantation, coupled with the differentiation of stem cells to facilitate the repair of damage. This approach holds promise for the treatment and potential cure of associated diseases [82]. Wang et al. prepared a hydrogel scaffold that can promote the growth of nerve axis [83]. They first generated poly(γ-benzyl-L-glutamate)-γ-poly-L-glutamic-acid radon copolymer and obtained the scaffold through an electrospinning technique. Finally, they evaluated the 3D scaffold using PC-3 differentiation and growth. Researchers have found that as the concentration of glutamate increases, the length of neural processes increases, indicating that glutamate has a promoting effect on the growth of neural. This also indicates that glutamate will play a positive role in the treatment of neurological-disorder-related diseases. For the study of its degradation performance, workers put the prepared scaffold in an environment of 37 °C, 5% CO_2_, immersed it in phosphate buffer, and found that its degradation was slow in 47 days. However, the internal environment is relatively complex, and whether the 3D scaffold can maintain stability in the body is still a problem that needs to be solved, Relatedly, the surface of electrospinning scaffolds is relatively hard, and cells may exhibit abnormal adhesion during the growth process; in contrast, a hydrogel scaffold becomes a good choice. At present, there is limited research on the application of γ-PGA in neural tissue engineering. The advantage of glutamate promoting the growth of neural processes also indicates that γ-PGA scaffolds may be better able to promote the recovery of neurological disorders than other scaffold materials.

### 6.3. Bone Tissue Engineering

Because of the irregular shape of bone defects, it is very difficult to construct and implant biological scaffolds, so hydrogels have broad prospects in bone tissue engineering [84]. Bo et al. constructed mussel-inspired bisphosphonated injectable nanocomposite hydrogels based on nHA, bisphosphonate–hydrazide-difunctionalized poly(l-glutamic acid) (PLGA-BP-ADH), and aldehyde–catechol-difunctionalized dextran (Dex-CHO-DP) (Figure 10C) [85]. Among them, nHA is the main component of bone matrix, which can be used to prepare composite hydrogels. The experiment proved that the hydrogel had good self repair performance and good adhesion. After it was implanted into the rat skull, the bone defect was reduced and new bone tissue was generated. Injectable hydrogels are prone to inflammatory reaction due to poor adhesion, and some implanted hydrogels are prone to deformation in load-bearing tissues, which may lead to the risk of infection. This kind of hydrogel has good self-healing property and adhesion ability, which overcomes some of the above limitations and is expected to become one of the candidate materials for bone tissue engineering in the future. The application of γ-PGA in tissue engineering is still in the research stage, and its true clinical application still requires extensive experimental verification.

**Figure 10 materials-17-00015-f010:**
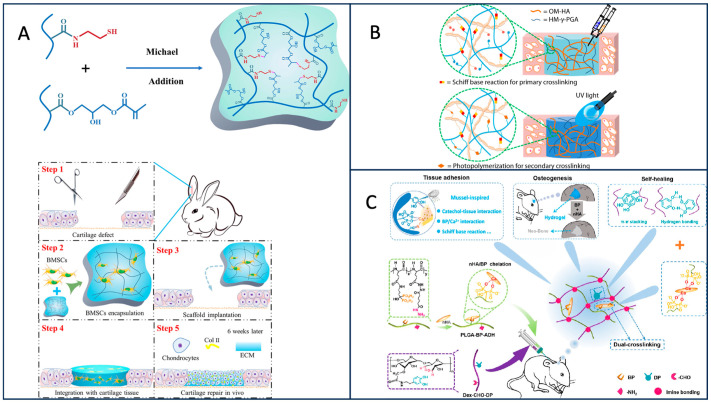
(**A**) Hydrogel formation mechanism and in vivo cartilage engineering diagram [76]. (**B**) Schematic diagram of single–double-crosslinked HA/γ-PGA hydrogel [79]. (**C**) Schematic diagram of fabrication of injectable bisphosphonated nanocomposite hydrogel inspired by mussels [85].

## 7. PGA for Wound Healing Applications

In wound healing, PGA-based dressings promote a moist wound environment, facilitate cell migration, and accelerate tissue regeneration. Functionalized PGA dressings with growth factors, antimicrobial agents, or other bioactive molecules further enhance wound healing capabilities, addressing a critical clinical need.

### 7.1. Antibacterial Effect of PGA in Wound Healing

In wound healing research, γ-PGA has demonstrated its antibacterial properties and potential to improve skin wound treatment. Several studies have explored the use of γ-PGA in wound dressings with antibacterial effects. Bae et al. found that PGA can promote corneal wound healing [86]. Shi et al. synthesized a γ-PGA/sericin (γ-PGA/SS) hydrogel and discovered its antibacterial properties [87]. Sun et al. modified ε-poly-lysine (ε-PL) and γ-PGA with glycidyl methacrylate (GMA) to create hydrogels with remarkable antibacterial effects against Staphylococcus aureus and Escherichia coli (Figure 11A) [88]. Sun et al. prepared an OKGM/γ-PGA DA-Cys/ε-PL (OKPP) hydrogel with antibacterial properties and wound healing effects, with experiments having shown that Pseudomonas aeruginosa and Staphylococcus aureus can be effectively inhibited by this hydrogel. This hydrogel also has antioxidant effects in vitro (Figure 11B) [89]. Liu et al. crosslinked γ-PGA and Pluronic F127 with a nitric oxide (NO) donor to form a hydrogel that inhibited bacterial growth while promoting wound healing. This combination of NO gas therapy can enhance the antibacterial effect of PGA, and studies have shown that both Escherichia coli and Staphylococcus aureus cultured in vitro can be inhibited by this hydrogel dressing (Figure 11F) [90]. Hu et al. developed a double-network hydrogel (CGLH) that incorporated epsilon-poly-lysine (epsilon-PL) and γ-PGA, resulting in enhanced antibacterial ability. Through clinical models, they demonstrated that CGLH could promote the healing of infected wounds and large wounds within 12 days and found that CGLH was beneficial to collagen synthesis and cell proliferation, suggesting that PGA might play a supporting role (Figure 11C) [91]. Wang et al. designed polymer vesicles (CIP-Ceria-PVs) containing ciprofloxacin (CIP) and cerium dioxide (Ceria). Since CIP-Ceria-PVs have activity similar to superoxide dismutase and thus can inhibit the formation of free radicals, it was found that CIP-Ceria-PVs have a more enhanced antibacterial effect than the CIP ratio through the diabetic wound model. Therefore, CIP-Ceria-PVs promote wound healing and have antioxidant and antibacterial effects (Figure 11D) [92]. Ramezani et al. utilized PGA’s biodegradable property to synthesize segmented polyurethane bound with polyglutamate peptide (PU-Pep) that degraded PGA through bacterial protease, preventing bacterial colonization and biofilm formation [93]. Tong et al. polymerized poly(γ-PGA)/alginate/Ag nanoparticles to create composite microspheres with excellent antibacterial hemostatic properties (Figure 11G) [94]. Yin et al. combined γ-PGA hydrogel, graphene oxide silver nanocomposite (GO-Ag), and multifunctional organic magnesium framework (Mg-mof) to promote wound healing in diabetic mice. PGA and Ag play a synergistic antibacterial role to further enhance the antibacterial effect. It was found that MN-MOF-GO-Ag can significantly improve the speed of wound healing through the skin wound model of diabetic mice (Figure 11E) [95]. These studies highlight the diverse potential applications of γ-PGA in wound healing, particularly its antibacterial properties, which are crucial for effective wound management. Further research is needed to explore the full extent of its capabilities and optimize its use in clinical settings.

**Figure 11 materials-17-00015-f011:**
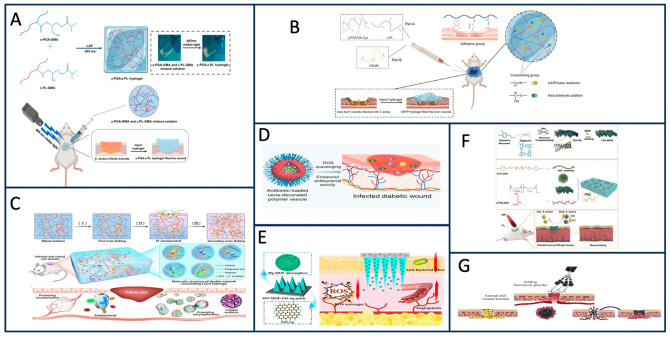
(**A**) Synthesis of γ-PGA-ε-PL hydrogel and treatment of infection model [88]. (**B**) This schematic shows the composition and source of adhesion properties of OKPP hydrogels [89]. (**C**) The preparation method of double-mesh CGLH and the schematic description of its biomedical application [91]. (**D**) Diagram of Ceria-modified vesicle loaded with antibiotics to heal diabetic wounds [92]. (**E**) Schematic diagram of MN-MOF-GO-Ag mechanism [95]. (**F**) Preparation of PDA-BNN6 nanosheets and hydrogels and application of hydrogels in wound healing of bacterial infection [90]. (**G**) Hemostatic agent coagulation process diagram [94].

### 7.2. Other Roles of PGA in Wound Healing

The γ-PGA’s unique crosslinked structure and high water absorption make it a valuable material for various applications, including wound healing and drug delivery. Wei et al. synthesized a super-absorbent hydrogel using PGA and ε-PL precursors, with casein as a blowing agent and calcium ion as a coagulant, addressing the low biocompatibility of ε-PL while retaining cell membrane penetration [96]. Further research is needed to optimize the PGA and ε-PL precursor ratios for optimal performance. Zhang et al. developed composite hydrogels comprising chitosan, heparin, and γ-PGA. Rheological experiments revealed the hydrogel’s remarkable swelling ability. To augment the antioxidant effect, superoxide dismutase was incorporated into the hydrogel. This hydrogel demonstrated accelerated wound healing in diabetic rat models, facilitating wound closure and collagen deposition. However, the dressing exhibited insufficient mechanical strength, impacting fibroblast migration (Figure 12A) [97]. Yang et al. engineered an injectable hydrogel with customizable mechanical strength by employing mercaptoylated γ-PGA and glycidyl methacrylate-conjugated oxidized hyaluronic acid to overcome the challenge of inadequate mechanical strength in wound dressings. The inclusion of mercaptan methacrylate (MMA) significantly enhanced the mechanical strength of γ-PGA-SH/OHA-GMA hydrogels. This augmented mechanical strength resulted in the spatial differentiation of fibroblasts, consequently expediting the process of wound healing (Figure 12B) [98]. γ-PGA can be grafted with other functional groups to create new materials. For example, Yang et al. designed a hydrogel combining collagen, hyaluronic acid, γ-PGA, and 3-aminophenylboric acid, which showed potential in improving the wound inflammatory microenvironment by promoting vascular proliferation [99]. Self-healing materials have been developed by modifying γ-PGA with SH and HA, creating a self-healing ECM-like hydrogel that promotes wound healing (Figure 12C) [100]. Xu et al. developed a multifunctional hydrogel fiber scaffold based on electrospinning and photocontrolled crosslinking γ-PGA/ginsenoside Rg^3^ (GS-Rg^3^) for preventing hypertrophic scars and clinical applications [101]. These studies demonstrate the versatility of γ-PGA in addressing various challenges in wound healing, including biocompatibility, mechanical strength, and drug delivery, while promoting efficient wound closure and improved tissue regeneration. Further research and optimization are ongoing to harness its full potential.

**Figure 12 materials-17-00015-f012:**
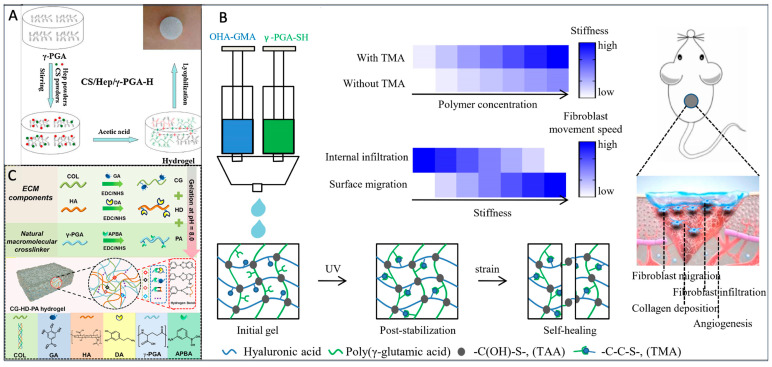
(**A**) Preparation of CS/Hep/γ-PGA composite hydrogels [97]. (**B**) Design, synthesis, and biological function diagram of γ-PGA-SH/OHA-GMA hydrogel [98]. (**C**) Synthesis of CG, HD, and PA and formation and construction mechanism of injectable self-healing CG-HD-Pa hydrogel [100].

## 8. Future Directions and Challenges

PGA demonstrates exceptional biocompatibility and biodegradability, underscoring its significance in various biomedical applications. The diverse preparation methods and applications of polyglutamate highlight its versatility, showcasing its potential impact on healthcare. From drug delivery to tissue engineering and wound healing, PGA-based materials consistently prove their importance in advancing biomedical technologies and enhancing patient care. This paper provides a comprehensive review of PGA’s roles and applications, including covalent modification with -COOH, surface charge modulation for encapsulating nanomaterials, and pH-responsive control of drug release.

Notably, PGA’s biodegradability, high water-retaining capacity, and amide bond-forming capabilities make it a valuable component for polyamide-PGA, surpassing hyaluronic acid in terms of superior moisturizing effects. The review emphasizes PGA’s potential in developing innovative biomedical materials, particularly within drug delivery systems and wound healing applications. Specific benefits discussed include the ability to modify degradation rates, enhance long-term stability, and achieve precise control over drug release kinetics. PGA’s versatility positions it as a promising candidate for shaping the future of biomedical materials and improving healthcare outcomes.

However, the translation of PGA-based materials from experimental stages to clinical applications faces several challenges: (1) complex and unstable production process: the production of medical-grade PGA materials is complex and often unstable, impacting the scalability of manufacturing; (2) lack of industry standards: the absence of standardized industry standards for medical-grade PGA leads to uncertainty in product usage, dosages, and potential side effects; and (3) clinical validation: PGA-based medical materials require extensive clinical data to establish their efficacy and safety. Considering these challenges, further research and development efforts are needed to optimize PGA materials for medical applications, enhance production processes, establish industry standards, and accumulate more clinical evidence supporting the efficacy of PGA-based medical materials. Overcoming these challenges will be essential to unlock PGA’s full potential in the biomedical field and improve its clinical adoption.

## Figures and Tables

**Figure 1 materials-17-00015-f001:**
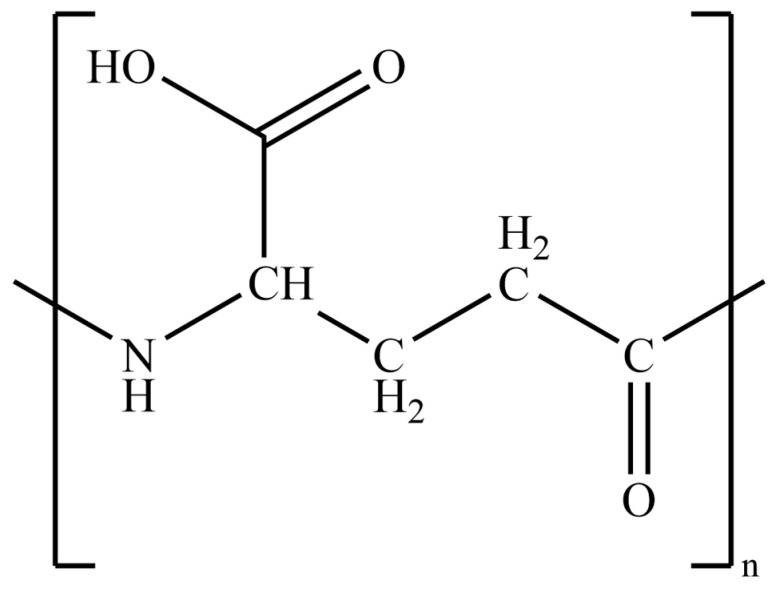
Molecular structure of poly-γ-glutamic acid.

**Figure 2 materials-17-00015-f002:**
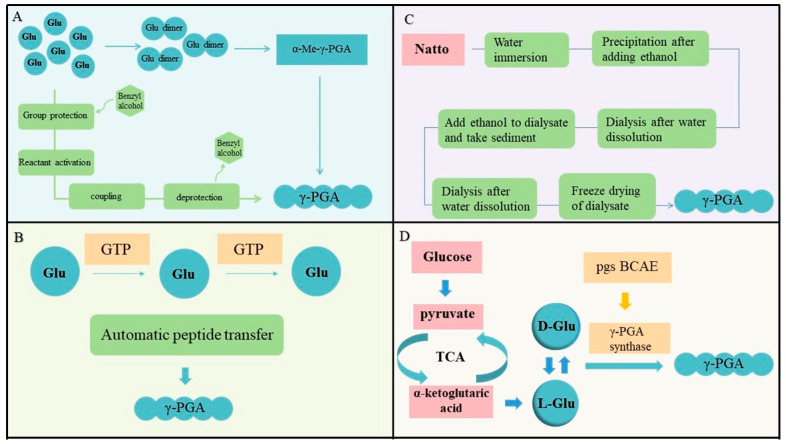
(**A**) The synthesis steps of traditional peptide synthesis method and dimer condensation polymerization method [9]. (**B**) The process of enzymatic conversion. GTP is glutamine transpeptidase [11]. (**C**) The process of extraction method. (**D**) The mechanism of microbial synthesis of γ-PGA.

**Figure 6 materials-17-00015-f006:**
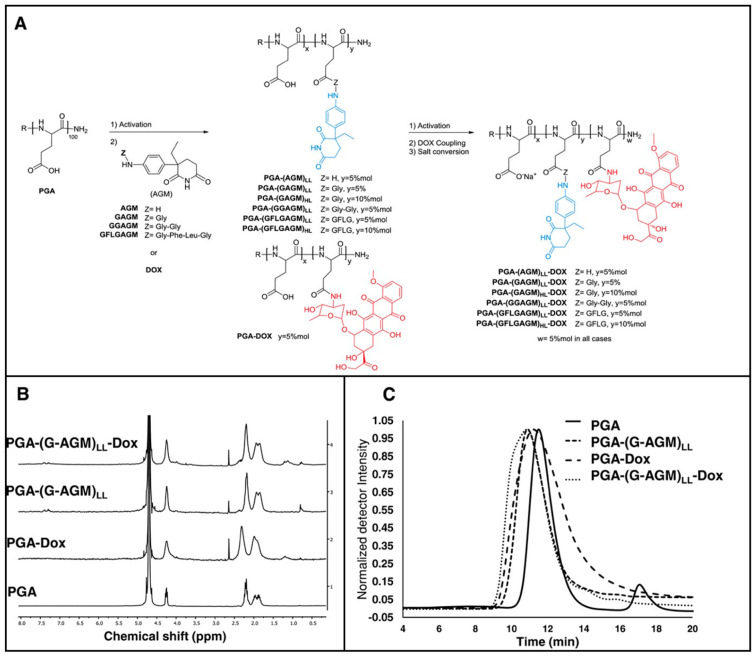
(**A**) The synthesis pathway for single-drug and combined-PGA-drug conjugates. (**B**) Representative 1H-NMR spectra recorded in D_2_O at 500 MHz. (**C**) A representative SEC chromatogram showing parental PGA and PGA-DOX, PGA-(G-AGM)LL, and PGA-(G-AGM)LL-DOX conjugates with UV detection at 260 nm [57].

**Table 1 materials-17-00015-t001:** Comparison of four PGA synthesis methods.

Synthesis Methods	Purity	Productive	Cost	Pollution	Applications
Chemical synthesis	Low	Low	High	High	Occasionally laboratory study
Enzyme conversion	High	High	High	Low	Soil, plant regulators, etc.
Extraction	Low	Low	Low	Low	Early production methods of γ-PGA
Microbial fermentation	High	Relatively high	Relatively low	Low	Commonly used for large-scale production of γ-PGA and applied in cosmetics, food, pharmaceuticals, water treatment, and other fields

## Data Availability

Data are contained within the article.

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
