# Peer review of "Synthesis of Poly-γ-Glutamic Acid and Its Application in Biomedical Materials"

_materials, 2023, doi:10.3390/ma17010015_

Round 1

Reviewer 1 Report

Comments and Suggestions for Authors

In this work, the authors have carefully described the polymer PGA underlining its versatility and potential impact on healthcare. It can be considered for publication in the Materials journal only if minor revisions are made that can better detail and explain some parts of the work, as highlighted below.

1) Table 1 line 83 change "synthesis methods to Synthesis methods", i.e. change the consonant s to a capital letter.

2) All figures, from n. 1 to n. 10, are not optimal in resolution, are illegible and not comprehensible. With the exception of figure n. 3 and figures n. 2 (C;D) all other figures are not mentioned in the text. Add.

Reviewer 2 Report

Comments and Suggestions for Authors

Dear Editor

Thank you. This article presented an interesting area of study. However, it is not clear to me whether the data presented in Fig 4 to Fig 12 has been performed by the author or is presented from others' work.  If not the author's won work then I recommend presenting their own work and minimizing the other's work. Please use others' work as citations only.

Best regards

Comments on the Quality of English Language

English needs improvement

Reviewer 3 Report

Comments and Suggestions for Authors

Dear authors,

I read your Review paper entitled "Synthesis of γ-PGA and Its Application in Biomedical Materials" and here are my comments:

1. Which is the added value of this review as there are similar papers in litearture on this topic. This must be explained in details for the field of biomedical applications. Also the methods to produce this polymer are extensively shown in literature.

2. Which is the research contribution of the authors in the field of γ-PGA (production, application)?

3. You could add some more applications in cardio-vascular system, and ophtlamology for example.

Reviewer 4 Report

Comments and Suggestions for Authors

This manuscript by Minjian Cai et al. reviews the sources of γ-poly(glutamic acid) and its medical or biomaterials applications.  The topic is interesting and although fairly well written, the manuscript contained a considerable number of minor issues.

I have only one scientific question.  γ-PGA contains a large number of carboxylate groups.  I presume these would produce a low pH in many biomaterials applications, such as tissue engineering and wound dressings.  Does that cause any problems and, if so, how is it alleviated?  I suggest some comments on this should be added to the manuscript.

Otherwise my comments concerns only minor typographical or stylistic issues.

Comments on the Quality of English Language

>  The introduction begins with an unnecessary tautology: 'at present' means the same as 'currently'.  One or other should be removed.

> When 'scientific' binomial names are used, the convention is to give the name in full (in italics) where it first appears, then to abbreviate the first part for subsequent mentions.  Hence, for example, Bacillus subtilis is given in full (in italics) where it first appears; subsequently, it should be abbreviated to B. subtilis.  

Note, the same is true where different species of the same genus are mentioned.  E.g. having mentioned Bacillus subtilis on L78, the names on L131 should be B. subtilis and B. licheniformis.

> A large number of abbreviations are used in the text.  The authors should ensure these are defined where they first appear.  In several cases, the definitions appeared later in the text.

> Some sentences were incomplete (words missing) and did not make sense.  E.G:

L427-429: 'The prepared hydrogel serves its application. Through characterization, select the appropriate hydrogel according to the application.'

L433-434: 'Especially in various types of cancer drug delivery has been widely concerned.'

> L509:  What is 'zymphoionic'?  Please explain.

> Miscellaneous typographical mistakes:

L412: 'of'

L438: Li et al.

L441: 'Singer'

L449: 'Nisar et al.'

Round 2

Reviewer 2 Report

Comments and Suggestions for Authors

Dear Editor

Thank you. Authors have improved their work in revised version. May be considered for publication.

Best regards

Md Abdus Subhan

Comments on the Quality of English Language

Dear Editor

Thank you. Authors have improved their work in revised version. May be considered for publication.

Best regards

Md Abdus Subhan